# Lung, Breast and Colorectal Cancer Incidence by Socioeconomic Status in Spain: A Population-Based Multilevel Study

**DOI:** 10.3390/cancers13112820

**Published:** 2021-06-05

**Authors:** Daniel Redondo-Sánchez, Rafael Marcos-Gragera, Marià Carulla, Arantza Lopez de Munain, Consol Sabater Gregori, Rosario Jimenez Chillarón, Marcela Guevara, Olivier Nuñez, Pablo Fernández-Navarro, María-José Sánchez, Miguel Angel Luque-Fernandez

**Affiliations:** 1Non-Communicable Disease and Cancer Epidemiology Group, Instituto de Investigación Biosanitaria de Granada, ibs.GRANADA, Calle Doctor Azpitarte 4, 18012 Granada, Spain; daniel.redondo.easp@juntadeandalucia.es (D.R.-S.); mariajose.sanchez.easp@juntadeandalucia.es (M.-J.S.); 2Consortium for Biomedical Research in Epidemiology and Public Health (CIBER Epidemiología y Salud Pública, CIBERESP), 28029 Madrid, Spain; rmarcos@iconcologia.net (R.M.-G.); mp.guevara.eslava@navarra.es (M.G.); onunez@isciii.es (O.N.); pfernandezn@isciii.es (P.F.-N.); 3Andalusian School of Public Health, Cuesta del Observatorio 4, 18080 Granada, Spain; 4Epidemiology Unit and Girona Cancer Registry, Oncology Coordination Plan, Department of Health, Autonomous Government of Catalonia, Catalan Institute of Oncology, Av. França, s/n, 17004 Girona, Spain; 5Descriptive Epidemiology, Genetics and Cancer Prevention Group, Biomedical Research Institute (IDIBGI), 17190 Girona, Spain; 6Tarragona Cancer Registry, Cancer Epidemiology and Prevention Service, Hospital Universitari Sant Joan de Reus, Av. Josep Laporte, 2, 43204 Reus, Spain; mcarulla@grupsagessa.com; 7Pere Virgili Health Research Institute (IISPV), Av. Josep Laporte, 2, 43204 Reus, Spain; 8Basque Country Cancer Registry, Health Department, Basque Government, Calle Donostia, 1, 01010 Vitoria-Gasteiz, Spain; arantza-lopez@euskadi.eus; 9Castellón Cancer Registry, Comunitat Valenciana Cancer Information System, Department of Health, Autonomous Government of Comunitat Valenciana, Av. De Cataluña, 21, 46020 València, Spain; sabater_congre@gva.es; 10Cuenca Cancer Registry, Department of Public Health, Autonomous Government of Castilla la Mancha, Calle de las Torres, 43, 16071 Cuenca, Spain; rjimenez@jccm.es; 11Navarra Public Health Institute, Calle Leyre, 15, 31003 Pamplona, Spain; 12IdiSNA, Navarra Institute for Health Research, Calle Irunlarrea, 3, 31008 Pamplona, Spain; 13Cancer and Environmental Epidemiology Unit, National Center for Epidemiology, National Institute of Health Carlos III, Av. Monforte de Lemos, 5, 28029 Madrid, Spain; 14Department of Preventive Medicine and Public Health, University of Granada, Av. De la Investigación, 11, 18071 Granada, Spain; 15Department of Noncommunicable Disease Epidemiology, ICON Group, London School of Hygiene and Tropical Medicine, Keppel Street, London WC1E 7HT, UK

**Keywords:** socioeconomic inequalities, colorectal cancer, lung cancer, breast cancer, epidemiology, population-based study

## Abstract

**Simple Summary:**

Despite political efforts across the world and Europe, social inequalities in cancer incidence are persistent. We studied the association between socioeconomic status (SES) and cancer incidence in nine Spanish provinces. Lower SES was associated with an increased risk of lung cancer among males. Higher SES was associated with an increased risk of breast cancer among females in Spain. Understanding the reasons behind the association between cancer incidence and SES could help develop appropriate public health programs to promote health and reduce socioeconomic inequalities in cancer incidence in Spain.

**Abstract:**

Socioeconomic inequalities in cancer incidence are not well documented in southern Europe. We aim to study the association between socioeconomic status (SES) and colorectal, lung, and breast cancer incidence in Spain. We conducted a multilevel study using data from Spanish population-based cancer registries, including incident cases diagnosed for the period 2010–2013 in nine Spanish provinces. We used Poisson mixed-effects models, including the census tract as a random intercept, to derive cancer incidence rate ratios by SES, adjusted for age and calendar year. Male adults with the lowest SES, compared to those with the highest SES, showed weak evidence of being at increased risk of lung cancer (risk ratio (RR): 1.18, 95% CI: 0.94–1.46) but showed moderate evidence of being at reduced risk of colorectal cancer (RR: 0.84, 95% CI: 0.74–0.97). Female adults with the lowest SES, compared to those with the highest SES, showed strong evidence of lower breast cancer incidence with 24% decreased risk (RR: 0.76, 95% CI: 0.68–0.85). Among females, we did not find evidence of an association between SES and lung or colorectal cancer. The associations found between SES and cancer incidence in Spain are consistent with those obtained in other European countries.

## 1. Introduction

The burden of cancer is rising globally, exerting a significant strain on populations and health systems at all income levels. Given the sociodemographic change in Western societies, cancer control will be one of the most complex health challenges in the future [1]. Estimates of population-based cancer incidence serve to evaluate cancer’s burden on health systems worldwide [2]. Despite political efforts across the world and Europe, social inequalities in cancer incidence are a persistent problem [3]. Social inequalities in cancer outcomes have an economic impact on healthcare costs [4]. Thus, identifying and characterizing socioeconomic and geographic disparities in cancer outcomes helps optimize and redistribute healthcare services in a more equitable fashion.

In northern Europe, there is a long-standing tradition of measuring deprivation in small geographical areas but not in Spain [5]. Recently, a standardized measure of socioeconomic deprivation covering Spain’s whole territory has been developed using the national census data from 2011, namely, the Spanish Deprivation Index (SDI). The index allows one to characterize and compare socioeconomic inequalities in cancer outcomes as a function of census tracts [6].

Using this newly developed index, we aim to study the association between socioeconomic inequalities and cancer incidence for three anatomical sites from the European High Resolution Studies [7] (i.e., colorectal, lung, and breast cancers) in Spain. Furthermore, we aim to investigate how cancer incidence varies geographically in small areas after accounting for age and the SDI in Spain during the period 2010–2013.

## 2. Materials and Methods

### 2.1. Study Design, Participants, Data, and Setting

We developed a population-based multilevel study. Data were drawn from nine Spanish population-based cancer registries (Albacete, Bizkaia, Castellón, Cuenca, Girona, Gipuzkoa, Granada, Navarra, and Tarragona) that participated in the European High Resolution Studies (TRANSCAN-HIGHCARE project within ERA-Net) [7]. Appendix A shows the location, within peninsular Spain, of the nine provinces of the study. Colorectal, lung, and breast cancer cases >18 years and diagnosed during the period 2010–2013 by census tract level, including their age, sex, and year of diagnosis, were included in the study. Case codes were C18–C21 with malignant behavior (/3) for colorectal, C34.0–C34.9 with malignant behavior (/3) for lung, and C50.0–C50.9 with malignant behavior (/3) or in situ (/2) for breast cancer among females according to the topography code of the International Classification of Diseases for Oncology, 3rd Edition [8]. Colorectal cancer data were used in a previous work to assess socioeconomic inequalities (measured with SDI) on survival in Spain [9].

Population-based figures broken down by census tract level, age, calendar year, and sex were obtained from the Spanish Statistical Office. The SDI was created by Duque et al. [6] using data from the Spanish 2011 census conducted by the Spanish National Statistics Institute. The index includes information from six indicators mainly related to employment and education: percentage of manual workers (employed or unemployed), percentage of occasional workers (employed or unemployed), percentage of the population with insufficient education, and percentage of main homes without internet access [6]. The index has no direct information about income, but in the sensitivity analysis, we found a direct association between SDI and average income per census tract [10] (Appendix A). We used the SDI divided in quintiles (Q), where Q5 represents the most deprived areas and Q1 the least deprived areas.

The internal review board of the Andalusian School of Public Health (CP17/00206) and the biomedical ethics committee of the Department of Health of the Andalusian Regional Government (study 0072-N-18) approved the study protocol.

### 2.2. Statistical Analysis

In the descriptive analysis, we first estimated the crude rates per 100,000 people for the overall period and by the quintiles of deprivation, categories of age, and sex. To compute the incidence rates, we used the total population at risk for the analyzed period. Afterward, we computed univariable- and multivariable-adjusted rate ratios using a Poisson mixed-effects model [11]. Models were stratified by sex. We added one variable at a time to control for confounding and used robust standard errors to account for overdispersion [12]. The final model was adjusted for SDI quintile, age in years, and calendar year. The model specification was given by
ln(*cases*/*population*) = β_0_ + β_1_ × (Quintile SDI) + β_2_ × (Age) + β_3_ × (Calendar year) + Q(1)
where Q is the random intercept for census tracts.

We included the census tract as a random intercept (Q in (1)) to account for spatial heterogeneity. From the models, we derived the Empirical Bayesian Estimate to identify patterns of non-random variation in cancer incidence in Spain [13]. Using the posterior Empirical Bayesian prediction, we mapped the age- and SES-standardized cancer incidence smoothed rates adjusted for the SDI by census tract in Spain for colorectal, lung, and breast cancers.

We assessed different model specifications in the sensitivity analysis, including the non-linearity effect of age using restricted cubic splines. Furthermore, we assessed the consistency of the reported cancer incidence rates by calendar year and cancer registry with published evidence in official statistical sources [14].

We used Stata v.16.1 (StataCorp, College Station, TX, USA) [15] and R v.4.0.2 (R Foundation for Statistical Computing, Vienna, Austria) [16] for statistical analysis.

## 3. Results

Figure 1 shows the SDI quintiles’ spatial distribution in 2011 by census tract for the nine Spanish provinces under study. Overall, there was a north–south pattern with the highest deprivation in the southern provinces (Granada, Albacete, and Cuenca) compared to those in the north of Spain (Bizkaia, Gipuzkoa, Girona, Navarra, and Tarragona). The province of Castellón showed an intermediate pattern between the northern and southern provinces. Appendix A shows the spatial distribution of the quintiles of SDI in the provinces’ capitals.

Table 1 shows the total number of cancer cases, the population at risk, and the observed incidence rates per 100,000 people during the period 2010–2013 by cancer site, sex, quintiles of deprivation, and age groups.

There were 3823 colorectal cancer cases and 6,147,118 people at risk in seven Spanish provinces during the period 2010–2013. Males showed a higher crude rate than females (77.4 per 100,000 males vs. 47.2 per 100,000 females). The crude incidence rate increased with age for both males and females, and, with quintiles of deprivation, only for females. Females with colorectal cancer living in the most deprived areas (i.e., Q5) showed 51.4 cases per 100,000 vs. 43.1 cases per 100,000 among females living in the least deprived areas (i.e., Q1).

There were 1067 lung cancer cases and 2,604,131 people at risk in two Spanish provinces during 2011–2012. Lung cancer risk increased with age for both males and females and with quintiles of deprivation only among males. Males with lung cancer living in the most deprived areas (i.e., Q5) showed a rate of 85.4 cases per 100,000 vs. 59.0 cases per 100,000 among males living in the least deprived areas (i.e., Q1).

There were 3157 breast cancer cases and 2,902,468 females at risk in six Spanish provinces during the period 2010–2013. Breast cancer incidence risk increased with age and decreased with deprivation levels (i.e., women living in the most deprived areas had a lower breast cancer incidence risk). Females with breast cancer living in the most deprived areas (i.e., Q5) showed a rate of 98.5 cases per 100,000 vs. 127.1 cases per 100,000 among females living in the least deprived areas (i.e., Q1).

Table 2 shows the cancer incidence risk by sex, adjusted for deprivation, age, calendar year, and accounting for the correlation within census tracts for colorectal, lung, and breast cancer during the period 2010–2013. After adjusting for age and calendar year (Model 3), the incidence risk of colorectal cancer among males living in the most deprived areas was 16% lower than for the males with colorectal cancer living in the least deprived areas (i.e., incidence rate ratio (IRR) Q5 vs. Q1: 0.84, 95% CI: 0.74–0.97). In females, there was no evidence of an association between deprivation and colorectal cancer incidence risk. There was weak evidence of an increased lung cancer incidence risk among the most deprived males (i.e., IRR Q5 vs. Q1: 1.18, 95% CI: 0.94–1.46) but not among women. For breast cancer, women from the least deprived areas showed an increased risk compared to women from the most deprived areas (i.e., IRR Q5 vs. Q1: 0.76, 95% CI: 0.68–0.85).

Figure 2, Figure 3 and Figure 4 show the observed and age- and SES-standardized incidence smoothed rates by census tracts for colorectal, lung, and breast cancers. The pattern of smoothed rates for colorectal cancer was characterized by a higher cancer incidence risk in northern Spanish provinces (i.e., Bizkaia, Girona, and Navarra) than in southern provinces (i.e., Granada and Albacete) (Figure 2). For lung cancer, Granada showed a higher risk than Girona (Figure 3). However, for breast cancer, the same north–south pattern was present, showing higher smoothed incidence rates in the northern provinces (i.e., Gipuzkoa, Navarra, Girona, and Tarragona) compared to the southern province of Granada (Figure 4). Furthermore, there was a north–south pattern regarding the spatial heterogeneity of the smoothed rates. The pattern was characterized by a higher heterogeneity of the rates in northern provinces than southern provinces except for lung cancer, where Granada showed a homogenous higher risk than Girona. Overall, for breast and colorectal cancer, the risk was higher in the census tracts from the province’s capital (Appendix A), while the opposite pattern was found for lung cancer (Appendix A).

Including the non-linear effect of age did not improve the overall model goodness of fit in the sensitivity analyses. Furthermore, the reported incidence rates were consistent with previously published evidence from official sources [17].

## 4. Discussion

We found that males with the lowest SES showed an increased risk of lung cancer incidence and a reduced risk of colorectal cancer incidence than those with the highest SES. Females with the lowest SES had a reduction in breast cancer incidence. Among women, we did not find evidence of an association between SES and lung or colorectal cancer.

Overall, our results are consistent with current evidence from population-based studies regarding cancer incidence and SES in Europe [18]. Low SES was associated with an increased risk of lung cancer in Denmark [19], England [20,21], France [22,23], Germany [24,25,26], Ireland [27], Italy [28], Slovenia [29], Sweden [30], and Scotland [31,32]. We found the same result in men, but the risk was not clear in women, perhaps due to the reduced number of cases. Furthermore, we found an inverse association between SES and breast cancer risk characterized by an increased risk among women with higher SES than those with lower SES. Likewise, the same pattern was found in England [21], France [22], Germany [24,25], Ireland [27], Italy [28], Scotland [32], and Slovenia [29]. However, there is a varying pattern of colorectal cancer incidence risk in Europe. Low SES was associated with an increased risk in England [21], Germany [25,26], and Scotland [31], while in other European countries, there was a reverse association or no association at all [18,23,24,27,29,33]. In our study, we found that males of more deprived areas were at a lower risk of colorectal cancer. We argue that it might be that there are two distinguished patterns of colorectal cancer risk in Europe related to lifestyle and cultural risk factors associated with SES [34]. However, more evidence is needed based on large periods and sample sizes. Furthermore, differences in the availability and impact of colorectal and breast cancer regional screening programs [35] might be associated with differences in the geographical distribution of cancer incidence rates.

### 4.1. Strengths

To the best of our knowledge, this is the first time that the association between area level SES and cancer incidence has been assessed in several Spanish provinces in a population-based study. Our study adds evidence regarding socioeconomic inequalities in cancer incidence around southern Europe. Furthermore, our results are consistent with other European countries and previous studies from Spain’s northeast region [36].

The number of incident cancer cases reported by calendar year for the majority of cancer registries was exhaustive. Appendix A show the distribution of the number of cases by anatomical site, year of diagnosis, and province. In the sensitivity analysis, we compared the reported incidence rates in our study by province and calendar year in the period 2011–2013 with official published statistics elsewhere [17]. Our study’s reported cancer incidence rates were consistent with the European Cancer Information System [17]. Even if there was no information from all the Spanish provinces in our study, most of the Spanish population-based cancer registries participated in the study, and the socioeconomic north–south gradient was well represented.

### 4.2. Limitations

Individual information on SES is often difficult to obtain due to ethical issues. Usually, it is not available at the population level, and cancer registries use census tracts and aggregated deprivation measures. SDI is not used as a proxy of individual SES but a measure of the contextual effect of living in a census tract with a specific SES. We highlight the importance of interpreting findings at the population level to minimize ecological fallacy risk. It has recently been shown that, in England and Wales, the average socioeconomic status at the area level correlates poorly with the individual SES in higher heterogeneity areas (i.e., ecological fallacy) [37]. Moreover, in our study, the SDI was derived for 2011, while cancer incidence is measured for the period 2010–2013. However, evidence shows that the aggregated measures of socioeconomic inequalities are consistent over time [38].

Finally, we suggest caution in interpreting the spatial patterns we identified. There is no significant random variability due to the short period under study and the limited number of cancer cases.

## 5. Conclusions

The study evaluates the association between socioeconomic inequalities and colorectal, lung, and breast cancer incidence in nine Spanish provinces during the period 2010–2013. The findings are consistent with other European countries, showing the association between socioeconomic inequalities and lung cancer among men and reduced breast cancer incidence among women with lower SES. Studying the causes of these associations could help develop appropriate public health programs to promote health and reduce socioeconomic inequalities in cancer incidence in Spain. Further collaborative studies are required to update the assessment of socioeconomic inequalities in colorectal, lung, and breast cancers over time in Spain and to evaluate public health programs.

## Figures and Tables

**Figure 1 cancers-13-02820-f001:**
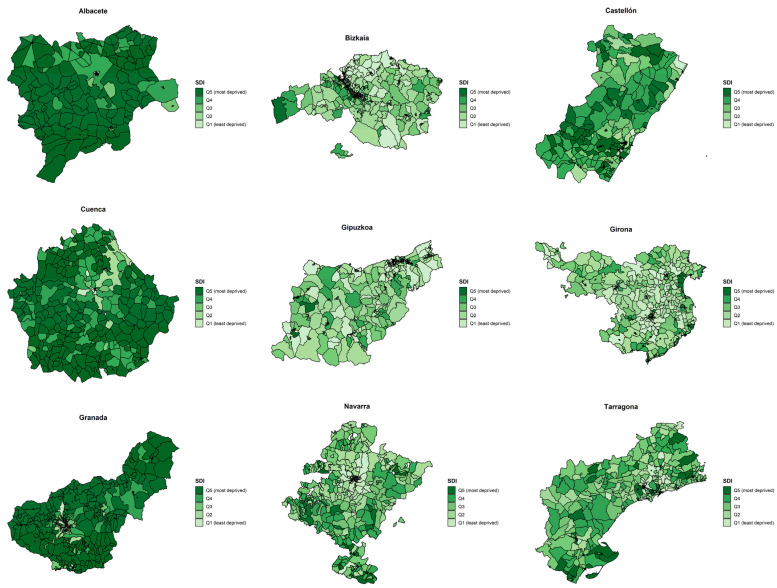
Quintiles of the Spanish Deprivation Index (SDI) by census tract in the nine Spanish provinces under study, 2011.

**Figure 2 cancers-13-02820-f002:**
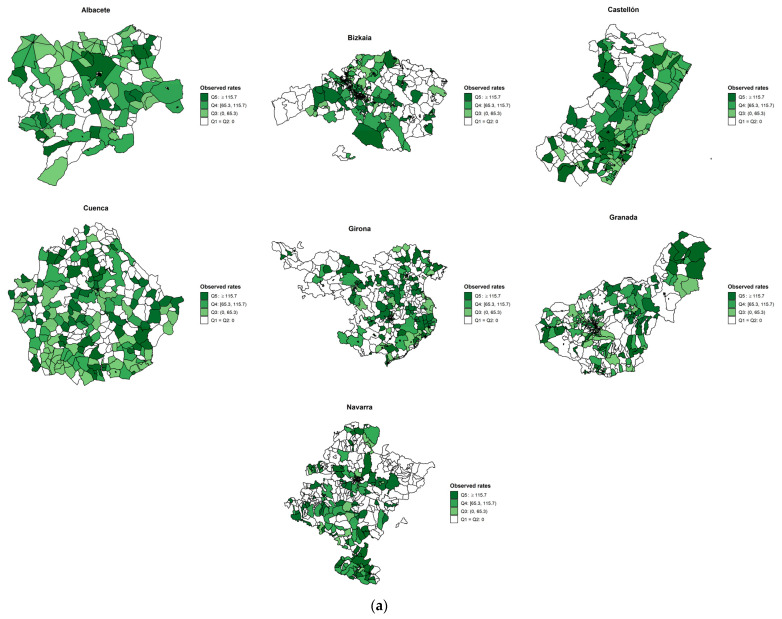
(**a**) Observed colorectal incidence rates and (**b**) smoothed cancer incidence rates adjusted for deprivation and age by census tract in seven Spanish provinces during the period 2010–2013.

**Figure 3 cancers-13-02820-f003:**
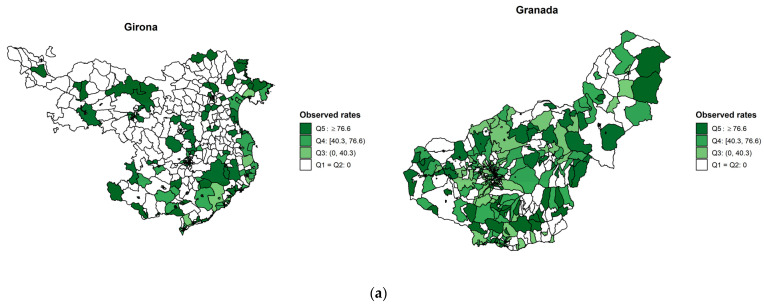
(**a**) Lung observed incidence rates and (**b**) smoothed cancer incidence rates adjusted for deprivation and age by census tract in two Spanish provinces during the period 2011–2012.

**Figure 4 cancers-13-02820-f004:**
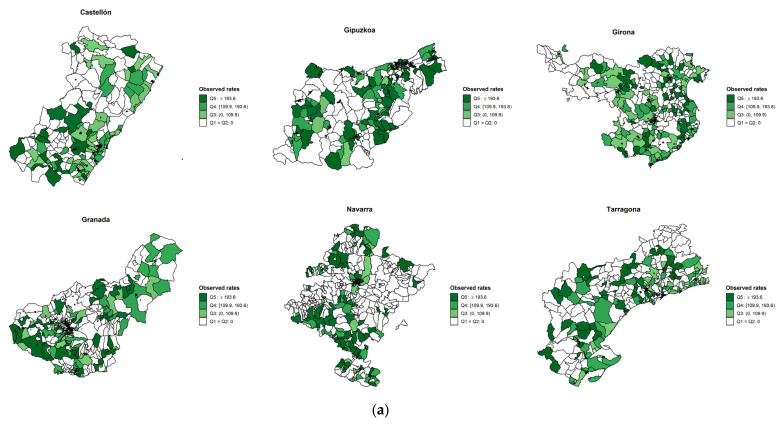
(**a**) Breast observed incidence rates and (**b**) smoothed cancer incidence rates adjusted for deprivation and age by census tract in seven Spanish provinces during the period 2010–2013.

**Table 1 cancers-13-02820-t001:** Cancer incidence, population at risk, incidence rate per 100,000 people, and *p*-value for trend by sex, deprivation, and age groups in Spain for the period 2010–2013.

	**Colorectal Cancer**							
	**Females**				**Males**			
Variables	Cases	Population at risk	Incidence rate * (95%CI)	*p*-value for trend	Cases	Population at risk	Incidence rate * (95%CI)	*p*-value for trend
Overall rate period 2010–2013	1457	3,088,569	47.2 (44.8–49.7)		2366	3,058,549	77.4 (74.3–80.5)	
Quintiles of SDI				0.031				0.288
Q1	306	710,049	43.1 (38.4–48.2)		502	666,960	75.2 (68.8–82.2)	
Q2	300	685,582	43.8 (39.0–49.0)		504	669,303	75.3 (68.9–82.2)	
Q3	341	676,128	50.4 (45.2–56.1)		500	673,987	74.2 (67.8–81.0)	
Q4	273	556,068	49.1 (43.4–55.3)		497	567,003	87.7 (80.1–95.7)	
Q5	237	460,742	51.4 (45.1–58.4)		363	481,296	75.4 (67.8–83.6)	
Age groups				<0.001				<0.001
<50	91	1,913,857	4.8 (38.3–58.4)		103	2,034,317	5.1 (4.1–6.1)	
50–54	89	209,507	42.5 (34.1–52.3)		110	215,299	51.1 (42.0–61.6)	
55–59	123	173,809	70.8 (58.8–84.4)		210	172,994	121.4 (105.5–139.0)	
60–64	131	161,485	81.1 (67.8–96.3)		280	155,412	180.2 (159.7–202.6)	
65–69	183	145,714	125.6 (108.1–145.2)		328	132,958	246.7 (220.7–274.9)	
70–74	161	129,505	124.3 (105.9–145.1)		337	109,816	306.9 (275.0–341.5)	
75–79	236	137,047	172.2 (150.9–195.7)		416	108,811	382.3 (346.5–420.9)	
80–84	240	112,979	212.4 (186.4–241.1)		354	77,129	459.0 (412.4–509.4)	
≥85	203	104,666	194.0 (168.2–222.5)		228	51,813	440.0 (384.8–501.0)	
	**Lung Cancer**							
	**Females**				**Males**			
Variables	Cases	Population at risk	Incidence rate * (95%CI)	*p*-value for trend	Cases	Population at risk	Incidence rate * (95%CI)	*p*-value for trend
Overall rate period 2011–2012	180	1,309,540	13.8 (11.8–15.9)		887	1,294,591	68.5 (64.1–73.2)	
Quintiles of SDI				0.555				0.004
Q1	39	231,526	16.8 (12.0–23.0)		125	211,780	59.0 (49.1–70.3)	
Q2	21	230,603	9.1 (5.6–13.9)		144	225,239	63.9 (53.9–75.3)	
Q3	47	272,080	17.3 (12.7–23.0)		171	270,082	63.6 (54.2–73.6)	
Q4	36	308,654	11.7 (8.2–16.2)		211	311,269	67.8 (59.0–77.6)	
Q5	37	266,677	13.9 (9.8–19.1)		236	276,221	85.4 (74.9–97.1)	
Age groups				<0.001				<0.001
<50	23	843,733	2.7 (1.7–4.1)		36	890,011	4.0 (2.8–5.6)	
50–54	34	88,734	38.3 (26.5–53.5)		50	88,958	56.2 (41.7–74.1)	
55–59	21	71,940	29.2 (18.1–44.6)		90	71,123	126.5 (101.8–155.5)	
60–64	26	65,445	39.7 (26.0–58.2)		99	61,409	161.2 (131.0–196.3)	
65–69	12	57,708	20.8 (10.7–36.3)		133	53,024	250.8 (210.0–297.3)	
70–74	7	50,656	13.8 (5.6–28.5)		138	42,560	324.3 (272.4–383,1)	
75–79	23	53,693	42.8 (27.2–64.3)		163	41,764	390.3 (332.7–455.0)	
80–84	17	41,883	40.6 (23.6–65.0)		122	27,786	439.1 (364.6–524.3)	
≥85	17	35,748	47.6 (27.7–76.1)		56	17,956	311.9 (235.6–405.0)	
	**Breast Cancer—Females**							
Variables	Cases	Population at risk	Incidence rate * (95%CI)	*p*-value for trend				
Overall rate period 2010–2013	3157	2,902,468	108.8 (105.0–112.6)					
Quintiles of deprivation				<0.001				
Q1	721	567,481	127.1 (118.0–136.7)					
Q2	631	558,370	113.0 (104.4–122.2)					
Q3	571	553,979	103.1 (94.8–111.9)					
Q4	629	608,179	103.4 (95.5–111.8)					
Q5	605	614,459	98.5 (90.8–106.6)					
Age groups				<0.001				
<50	912	1,839,937	49.6 (46.4–52.9)					
50–54	393	193,301	203.3 (183.7–224.4)					
55–59	332	165,866	200.2 (179.2–222.9)					
60–64	350	153,247	228.4 (205.1–253.6)					
65–69	323	133,590	241.8 (216.1–269.6)					
70–74	208	112,291	185.2 (160.9–212.2)					
75–79	263	117,201	224.4 (198.1–253.2)					
80–84	197	95,765	205.7 (178.0–236.5)					
≥85	179	91,270	196.1 (168.4–227.1)					

* Per 100,000 people.

**Table 2 cancers-13-02820-t002:** Cancer incidence rates ratios by sex, adjusted for deprivation and age in Spain for the period 2010–2013.

	**Colorectal Cancer**					
	**Females**			**Males**		
Variables	Model 1	Model 2	Model 3	Model 1	Model 2	Model 3
	IRR (95% CI)	IRR (95% CI)	IRR (95% CI)	IRR (95% CI)	IRR (95% CI)	IRR (95% CI)
Quintiles of SDI						
Q1	Ref.	Ref.	Ref.	Ref.	Ref.	Ref.
Q2	1.02 (0.87–1.20)	1.02 (0.87–1.20)	0.99 (0.84–1.16)	0.98 (0.86–1.12)	0.94 (0.83–1.07)	0.94 (0.83–1.07)
Q3	1.14 (0.98–1.34)	1.15 (0.98–1.34)	1.09 (0.93–1.27)	0.99 (0.87–1.13)	0.93 (0.82–1.05)	0.93 (0.82–1.05)
Q4	1.13 (0.97–1.33)	1.14 (0.97–1.34)	1.04 (0.89–1.22)	1.16 (1.01–1.32)	1.02 (0.90–1.17)	1.03 (0.91–1.18)
Q5	1.16 (0.98–1.38)	1.17 (0.98–1.39)	1.02 (0.86–1.22)	0.99 (0.86–1.14)	0.83 (0.72–0.95)	0.84 (0.74–0.97)
Age in years						
Per ten years increase		1.02 (0.95–1.10)	1.03 (0.96–1.11)		1.86 (1.81–1.92)	1.86 (1.81–1.92)
Period 2010–2013						
Per one year increase			1.58 (1.53–1.63)			1.10 (1.04–1.16)
	**Lung Cancer**					
	**Females**			**Males**		
Variables	Model 1	Model 2	Model 3	Model 1	Model 2	Model 3
	IRR (95% CI)	IRR (95% CI)	IRR (95% CI)	IRR (95% CI)	IRR (95% CI)	IRR (95% CI)
Quintiles of SDI						
Q1	Ref.	Ref.	Ref.	Ref.	Ref.	Ref.
Q2	0.56 (0.32–0.97)	0.56 (0.32–0.96)	0.57 (0.33–0.98)	1.06 (0.83–1.36)	1.08 (0.85–1.38)	1.08 (0.85–1.37)
Q3	1.02 (0.66–1.57)	1.01 (0.66–1.56)	1.03 (0.67–1.58)	1.04 (0.83–1.30)	1.07 (0.85–1.34)	1.07 (0.85–1.34)
Q4	0.71 (0.45–1.13)	0.72 (0.45–1.15)	0.72 (0.45–1.14)	1.14 (0.91–1.42)	1.12 (0.90–1.39)	1.12 (0.90–1.39)
Q5	0.80 (0.50–1.28)	0.82 (0.51–1.31)	0.76 (0.47–1.22)	1.37 (1.09–1.71)	1.17 (0.94–1.46)	1.18 (0.94–1.46)
Age in years						
Per ten years increase		0.86 (0.62–1.18)	0.85 (0.62–1.17)		2.03 (1.94–2.13)	2.03 (1.94–2.13)
Period 2011–2012						
Per one year increase			1.50 (1.40–1.61)			1.00 (0.87–1.15)
	**Breast Cancer—Females**					
Variables	Model 1	Model 2	Model 3			
	IRR (95% CI)	IRR (95% CI)	IRR (95% CI)			
Quintiles of SDI						
Q1	Ref.	Ref.	Ref.			
Q2	0.90 (0.80–1.00)	0.91 (0.81–1.01)	0.89 (0.80–0.99)			
Q3	0.83 (0.74–0.93)	0.84 (0.75–0.94)	0.81 (0.73–0.91)			
Q4	0.82 (0.74–0.92)	0.83 (0.74–0.92)	0.79 (0.71–0.89)			
Q5	0.79 (0.71–0.88)	0.80 (0.72–0.89)	0.76 (0.68–0.85)			
Age in years						
Per ten years increase		1.07 (1.03–1.12)	1.06 (1.02–1.11)			
Period 2010–2013						
Per one year increase			1.42 (1.40–1.44)			

IRR: Incidence rate ratio. Model 1: model adjusted for quintiles of the SDI. Model 2: model adjusted for quintiles of the SDI and age. Model 3: model adjusted for quintiles of the SDI, age, and year of diagnosis.

## Data Availability

This research has been conducted using the Spanish National Statistics Institute data under application number BE181/2018 granting access to the corresponding population and mortality data. De-identified participant data from the Spanish cancer registries data are available via the High Resolution Studies on application. Proposals should be directed to the Spanish National Statistics Institute (https://www.ine.es/, accessed on 27 May 2021) and the High Resolution Studies (http://www.hrstudies.eu/contacts.html, accessed on 27 May 2021) to gain access; data requestors will need to sign a data access agreement.

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
