# Peer review of "Lung, Breast and Colorectal Cancer Incidence by Socioeconomic Status in Spain: A Population-Based Multilevel Study"

_cancers, 2021, doi:10.3390/cancers13112820_

Round 1
Reviewer 1 Report
This is a very good research and well design manuscript. The authors use a Spanish deprivation index (SDI) as surrogate of social inequality, and compare the cancer incidence between different degree social economic status among several regions in Spain for the colorectal cancer, breast cancer and lung cancer. These three cancers are highly related to socio-economic status among low socioeconomic population might have higher prevalence of cigarette smoking, environmental pollution and poor nutrition parents. The results are good correlate with the status of Europe, and the proved SDI is a good surrogate indicator for social economic inequalities. I strongly suggest this paper be published in this journal.
Author Response
Thank you for your comment, we appreciate your time and effort.
Reviewer 2 Report
The design of the study appears to be interesting and is well structured and discussed, also the results, interesting, appear solid and well described.
Author Response

(The authors gave the same response as above.)

Reviewer 3 Report
This is a good paper and I don't have any major comments. Congrats. Can the authors clarify why lung cancer was observed over a shorter period?
Author Response
Thank you for your comment, we appreciate your time and effort. The shorter period observed for lung cancer was related to the way the participants cancer registries provided the information to the European collaboration TRANSCAN-HIGHCARE project within ERA-Net. Cancer registries reported cancer incidence in different years from 2010 to 2013, but only two cancer registries reported lung cancer incident data from 2011 and 2012.
Reviewer 4 Report
In this study, the author investigated the association between socioeconomic status and colorectal, lung, and breast cancer incidence in nine Spanish provinces during 2010-2013. They found that lower socioeconomic status was associated with an increased risk of lung cancer among males. Higher socioeconomic status was associated with an increased risk of breast cancer among females in Spain.
There are some issues required to be addressed. The specific comments are listed below:
- I found some of the data has been published. The colorectal cancer part is similar with published paper “Socioeconomic Inequalities in Colorectal Cancer Survival in Southern Spain: A Multilevel Population-Based Cohort Study” (PMID: 32801917). The author should delete the repeating parts, or appropriate reference to previous article.
- The data is from 2010-2013. It is a little old. Try to find some new data.
- Figure 2-4 are too small to read.
- There are some spelling errors and abbreviations misuse, the author should check it carefully.
Round 2
Reviewer 4 Report
The revised version is improved.